# Analytical Performances of the Novel i-STAT Alinity Point-of-Care Analyzer

**DOI:** 10.3390/diagnostics13020297

**Published:** 2023-01-13

**Authors:** Romaric Larcher, Maxence Lottelier, Stephanie Badiou, Anne-Marie Dupuy, Anne-Sophie Bargnoux, Jean-Paul Cristol

**Affiliations:** 1Biochemistry and Hormonology Department, Montpellier University Hospital, PhyMedExp (Physiology and Experimental Medicine), INSERM (French Institute of Health and Medical Research), CNRS (French National Centre for Scientific Research), University of Montpellier, 34000 Montpellier, France; 2Infectious and Tropical Diseases Department, Nimes University Hospital, 30000 Nimes, France; 3Biochemistry and Hormonology Department, Montpellier University Hospital, 34000 Montpellier, France

**Keywords:** analytical performance, Point-of-Care Analyzer, i-STAT Alinity, blood gas, lactate, chemistry, urea, creatinine, hematology, coagulation

## Abstract

Many Point-of-Care devices have been released over the past decade. However, data regarding their analytical performances in real-world situations remains scarce. Herein, we aimed to assess the analytical performances of the i-STAT Alinity system. We conducted an analytical performances study with the i-STAT Alinity device using cartridges CG4+ (pH, Pco_2_, Po_2_, lactate, bicarbonate and base excess); CHEM8+ (Na, K, Cl, ionized Ca, urea, creatinine, glucose, hematocrit and hemoglobin) and PT/INR (prothrombin time and international normalized ratio). We assessed the imprecision and compared the results to those obtained on existing instruments in the central laboratory. We found that the within-lab coefficients of variation (CV) were very low (<2%) or low (2–5%), except for creatinine and PT (CV = 5.2% and CV = 6.3%, respectively). For almost all the parameters, the results were strongly (R^2^ = 90–95%) or very strongly (R^2^ > 95%) correlated with those of the existing laboratory instruments, and the biases were very low (<2%) or low (2–5%). However, correlations of the PT and INR measurements with existing instruments were lower (R^2^ = 86.0% and 89.7%), and biases in the Po_2_ (7.9%), creatinine (5.4%) and PT (−6.6%) measurements were higher. The i-STAT Alinity appeared as a convenient device for measurements of numerous parameters. However, clinicians should interpret Po_2_, creatinine and PT results with caution.

## 1. Introduction

Point-of-Care (POC) testing is defined as testing performed near or at a patient’s bedside instead of in a conventional central laboratory [1]. POC testing results are available rapidly, helping physicians in diagnosis and the prompt initiation and monitoring of therapeutic interventions [1]. In the last decades, POC testing applications have increased, especially in Emergency Departments (EDs) and Intensive Care Units (ICUs), where rapid diagnosis and treatment are a cornerstone of patient care [1,2,3].

Thanks to advances in technologies and the miniaturization of electronic devices, just in minutes, novel POC analyzers are now able to measure a large number of parameters from a small blood sample. Indeed, beyond a simple blood gas analysis with basic chemistry and hematology, measurements such as renal, cardiac or inflammatory biomarkers and coagulation tests are now available [4,5,6,7]. The limitations of the turnaround times of these investigations hold promise to change patients’ care with acute coronary syndromes [3], hemorrhage [5], sepsis [8] or metabolic disorders [9], among others. 

However, most of the POC users are clinicians, nurses or midwives with limited technical training. Therefore, biologists should assess POC analyzer precision remains equivalent to those obtained in central laboratories to ensure the proper diagnosis, management of patients and concordance of the threshold values for therapeutic decision making [4]. Thus, analytical performance studies in real-world situations are mandatory for novel POC devices. Given the large number of POC systems available, and the large number of biological parameters measured, data are scarce in the literature, especially on novel POC devices such as the i-STAT Alinity system (Abbott laboratories, Chicago, IL, USA) [4,5].

In this study, we aimed to assess the analytical performances of the i-STAT Alinity system simultaneously for a large panel of analyses, including blood gas, lactate, sodium (Na), potassium (K), ionized calcium (iCa), glucose, urea, creatinine, hemoglobin (Hb), hematocrit (Ht), prothrombin time (PT) and international normalized ratio (INR). Moreover, we aimed to assess the ability of the i-STAT Alinity device to be used for the management of vitamin K antagonist (VKA) therapy and the classification of patients with chronic kidney disease (CKD).

## 2. Materials and Methods

### 2.1. Study Design and Setting

We carried out the prospective analytical performance study of the i-STAT Alinity POC device in the Biochemistry and Hematology Laboratories of Montpellier University Hospital, France, between August, 2019 and February, 2020 using the following cartridges: CG4+ (pH, partial pressure of carbon dioxide or Pco_2_ and partial pressure of oxygen or Po_2_ and lactate); CHEM8+ (Na, K, Cl, iCa, urea, creatinine, glucose and Ht/Hb) and PT/INR (PT and INR). During the study period, whole blood samples drawn from inpatients were randomly screened after routine blood gas analyses or PT/INR measurements. A single operator (M.L.), trained in the use of the i-STAT Alinity system, performed the analyses. 

### 2.2. Patient Specimens

Whole blood samples obtained from the patient were collected in a heparinized syringe (safePICO syringe, Radiometer, Copenhagen, Denmark) for both the laboratory blood gas and the i-STAT measurements (except for coagulation). Lithium heparin, sodium citrate and no additive (neutral) blood collection tubes (BD Vacutainer™, Becton Dickinson, Plymouth, UK) were used, respectively, for the urea and creatinine measurements in the central laboratory, PT and INR measurements in the central laboratory and for PT and INR measurements on the i-STAT device.

### 2.3. Point-Of-Care Testing: i-STAT Alinity System

The i-STAT Alinity system is an easy-to-use portable blood analyzer that delivers test results in a few minutes. This POC system uses a single-use disposable cartridge containing microsensors, a calibrator and a waste container. The calibration solution is stored in a sealed cartridge and automatically released to perform a one-point calibration just before the sample analysis. The entire fluid path is also stored in the sealed cartridge and requires less than 100 microliters of whole blood introduced either from a heparinized blood gas syringe, a capillary or a neutral tube with a tip applicator. In this study, we focused on 3 of the 17 cartridges available: (1) the i-STAT CG4+ cartridge measuring pH and Pco_2_ by direct potentiometry; Po_2_ and lactate by amperometry and calculating the bicarbonate, base excess, total CO_2_ (Tco_2_) and O_2_ saturation (So_2_) in two minutes; (2) the i-STAT CHEM8+ cartridge measuring Na, K, Cl, iCa, Tco_2_ and urea by ion-selective electrode potentiometry, creatinine and glucose by amperometry after the enzymatic reaction; hematocrit by conductometry and calculating the anion gap and hemoglobin in two minutes and (3) the i-STAT PT/INR cartridge measuring PT and INR by amperometry after enzymatic reactions in five minutes. 

### 2.4. Imprecision Study

The imprecision of this cartridge-type POC was tested through the measurement of two levels of control material (Abbott i-STAT TriControl) over the course of one day at one site, by the same operator (M.L.) with one lot of test cartridges. The internal precision study was performed with duplicates of each sample tested five times a day on 2 i-STAT Alinity devices.

### 2.5. Comparison Study

Anonymized whole blood samples were tested on the POC i-STAT Alinity instrument using one lot of cartridges: CG4+, CHEM8+ and PT/INR. The i-STAT Alinity system results were compared to those obtained on existing instruments in central laboratories. The results for pH, Pco_2_, Po_2_, lactate, sodium, potassium, chlorine, ionized calcium, glucose, hemoglobin and hematocrit were compared to those obtained using the ABL90 Flex Plus blood gas analyzer (Radiometer, Copenhagen, Denmark). The results for urea and creatinine were compared to those obtained by enzymatic methods on a Cobas 8000 instrument using a c702 module (Roche Diagnostic, Meylan, France). The PT and INR measurements were compared to those determined using an ACL TOP 700 analyzer (Werfen, Barcelona, Spain).

### 2.6. Estimation of Glomerular Filtration Rates and Management of Vitamin K Antagonists

The Chronic Kidney Disease Epidemiology Collaboration (CKD-EPI) formula was used to calculate the estimated glomerular filtration rate (eGFR), and patients were classified according to the Kidney Disease: Improving Global Outcomes (KDIGO) guidelines [10].

An INR value between 2.0 and 3.0 was considered the target for patients treated with VKA according to the 2020 European Society of Cardiology (ESC) Guidelines for the diagnosis and management of atrial fibrillation [11].

### 2.7. Statistical Analysis

The means, standard deviations (SD) and coefficients of variation (CV) for each level of control were determined in the imprecision study. Multiple sources of analytical performance goals were considered for comparison purposes [12]. The database of Ricos et al. [13,14] was used preferably; however, some parameters were not available in this database (Po_2_) or were evaluated in an old study (pH). The Royal College of Pathologists of Australasia (RCPA) database [15] for the total error and the French Society of Clinical Biology (SFBC: Societe Francaise de Biologie Clinique) database [16] for imprecision goals were considered in these cases.

The Passing–Bablok regression analysis was used to compare the results of the POC i-STAT Alinity instrument with those obtained with existing laboratory instruments. The scatter of differences was visualized by means of Bland–Altman plots [17]. 

Agreement between the analyzers for CKD staging based on creatinine values was assessed by Cohen’s kappa test. Taking into account that CKD stages are ordered and for how far apart the two analyzer measurements are, a weighted kappa was calculated [18]. Cohen’s kappa results were interpreted as follows: 0.00–0.20 as none to slight, 0.21–0.40 as fair, 0.41–0.60 as moderate, 0.61–0.80 as substantial and 0.81–1.00 as almost perfect agreement. 

The same analysis was conducted to assess the agreement between the analyzers for the management of patients treated with VKA based on INR values.

Data analysis and statistical calculations were performed using XLSTAT^®^ software (version 2016.06.35661, Microsoft, New York, NY, USA).

## 3. Results

### 3.1. Imprecision Study

The imprecision study results are summarized in Table 1. 

The within-lab CVs were very low (≤2.0%) or low (2.0–5.0%), except for low values of creatinine and high values of PT, in which the CVs were at 5.2% and 6.3%, respectively. In addition, the CV values for these parameters were above the desirable values expected for the central laboratory instruments.

### 3.2. Comparison Study

We analyzed excess whole blood from 49 patients (30 males and 19 females, median age: 69 years old (interquartile range: 55–80)) collected in a heparinized syringe and 45 serum samples in lithium heparin collection tubes. 

In addition, 46 blood samples collected in no additive collection tubes (neutral purge tubes) and citrate collection tubes were taken from 46 patients (28 males and 18 females, median age: 67 years old (interquartile range: 61–75)).

The results of the Passing–Bablok regression and Bland–Altman studies are reported in Table 2 and Figure 1.

As illustrated in Figure 1, the i-STAT Alinity tests were strongly (R^2^ = 90–95%) or very strongly (R^2^ > 95%) correlated with the existing laboratory instrument measurements, and the biases were very low (<2.0%) or low (2.0–5.0%) for almost all the parameters. However, the PT measurements and INR calculations with the iSTAT Alinity instrument showed a lower correlation with the existing laboratory instrument measurements: R^2^ at 86.0% and 89.7%, respectively (Figure 1). Biases at 7.9%, 5.4% and −6.6% for the Po_2_, creatinine and PT determination, respectively, were observed (Table 2).

### 3.3. Agreement between Devices in Clinical Practice

The calculation of eGFR was done in 45 patients with creatinine values obtained with the POC i-STAT Alinity and the Cobas 8000 instruments. Then, patients were classified according to their CKD stage (Table 3).

The correlation of classification for patients with eGFR ≥ 60 mL/min/1.73m^2^ was perfect. However, due to bias in the creatinine measurements, seven patients with eGFR < 60 mL/min/1.73m^2^ were classified in a different stage (more severe with the i-STAT Alinity device). The percentage of agreement between the Cobas 8000 and i-STAT Alinity for CKD staging was 84.4%, with a crude Cohen’s kappa at 0.807 (95% confidence interval from 0.679 to 0.934). Given that misclassified patients only had one stage difference (three were classified IIIa/IIIb, two IIIb/IV and two IV/V), the weighted kappa was estimated at 0.926.

A comparison of the INR results obtained with the POC i-STAT Alinity and the ACT TOP 700 instruments in 46 patients showed that only four of them would have had different management for their VKA treatment (Table 4).

The percentage of agreement between the ACT TOP 700 and the i-STAT Alinity for management of the VKA treatment was 84.4%, with a crude Cohen’s kappa at 0.831 (95% confidence interval from 0.675 to 0.986) and a weighted kappa estimated at 0.850. 

These results underlined the good agreements between the devices for CKD staging and the management of VKA treatment in clinical settings.

## 4. Discussion

We reported herein the results of an analytical performances study of the i-STAT Alinity system for blood gas, lactate, electrolytes, glucose, urea, creatinine, hematocrit, hemoglobin and PT/INR measurements. Overall, the analytical performances in real-world situations were in agreement with the manufacturer specifications, including the creatinine and INR levels. In addition, POC testing with i-STAT Alinity reached desirable imprecision in the measurement of almost all parameters, and almost all these measurements were well corelated with the existing instruments in central laboratories. Nevertheless, the Po_2_, creatinine and PT/INR measurements were associated with higher imprecision and/or biases in comparison with the existing methods. However, the clinical impact in patients with hypoxemia, CKD or treated with VKA seemed limited.

A growing number of POC analyzers are released and offer increasingly wide analysis panels at the bedside instead of in a conventional central laboratory. Among them, the i-STAT Alinity system is an easy-to-use portable blood analyzer allowing the performance of rapid diagnostics at the POC in a hospital, or even home, setting. Unsurprisingly, the use of POC testing is increasing [1,2,3]; however, the analytical performance goals for POC analyzers have not yet been published and remain unclear. Nonetheless, it can be expected that the analytical performances of POC testing should meet the requirements regarding bias and precision as for measurements performed on analyzers in the central laboratory [13,14,16]. 

In order to compare the analytical performances of the i-STAT Alinity with the other systems, we performed a deep review of the literature on similar commercial devices (see Appendix A). This review clearly showed that i-STAT Alinity is one of the most complete apparatuses that could perform, using the same device, a blood gas analysis, ionic disturbances monitoring, kidney function evaluation and VKA follow-up. Morevover, the analytical performance of i-STAT Alinity was in the range of other systems. Thus, we could assess the performance of each parameter mesearument, compare them with the central lab results and appreciate the clinical concordance.

Our results are in accordance with previous studies that reported very good and good precision and correlation with the existing methods for pH, Pco_2_, bicarbonate, lactate, electrolytes, glucose, hematocrit and hemoglobin tests [1,4,19]. In our imprecision study, the CVs of these parameters were also in accordance with the analytical performance goals [13,16]. On the contrary, the CVs of the Po_2_ and creatinine (values < 80 µmol/L) measurements were above the goals defined by the SFBC [16] but were lower than those previously reported [1,4,20], except for one study that reported a CV at 1.4% for Po_2_ [1].

Most importantly, we found a positive average bias (5.4%) in the creatinine tests versus the Roche isotopic dilution mass spectrometry (IDMS) calibrated enzymatic assay performed in the central lab. As previously described [21], this was predominantly at higher creatinine levels above 100 µmol/L, leading to an overestimation of more than 20 µmol/L for the highest values of creatinine concentration. However, the clinical impact regarding the staging of CKD in real-life settings seems acceptable, since the agreement between devices was excellent. Interestingly, no misclassification was observed for patients with eGFR ≥ 60 mL/min/1.73 m^2^, which is generally assumed as an alert threshold, in particular, for contrast nephropathy and numerous drug dosage regimen adjustments. This point should be highlighted, since POC testing devices are intended for EDs and imaging departments in order to fluidify patient care rather than for nephrology departments or dialysis centers.

We also reported a 7.9% bias in the Po_2_ measurements that was not reported in other studies evaluating the analytical performances of i-STAT Alinity [1,4,19]. This bias could be related to preanalytical factors such as time between analyses on the different analyzers and/or may be attributed to the different sensor systems that react differently towards the same patient matrix [22]. Interestingly, this bias in the Po_2_ measurements mainly impacted the highest value of Po_2_ (above 150 mmHg), which limited its clinical importance [23,24] and has been observed with other blood gas analyzers [12]. In the same vein, we found a bias of −7.3% in the hematocrit measurement, which was already underlined in one study [4] but not confirmed in another [1], and whose clinical impact seems likely limited [4]. On the contrary, although i-STAT Alinity seemed to be a reliable POC device for iCa measurements, it should be specifically evaluated in ICU settings for monitoring continuous venovenous hemodialysis with regional citrate anticoagulation to ensure that it does not induce undue changes in the citrate dose [25,26].

To the best of our knowledge, this study was the first to analyze the performance of i-STAT Alinity for PT/INR testing. Schober et al. [5] recently reported that a blood analysis with the i-STAT-1 POC PT/INR coagulation measurement was feasible in a helicopter emergency medical service in a cohort of 15 patients. However, the analytical performance of the PT/INR test was not assessed. Using i-STAT Alinity, we reported an imprecision above the standards expected for PT measurements [13] and an unsatisfactory bias at −6.6%. Nonetheless, we also highlighted that the imprecision and biases in PT and INR probably have limited impacts on the management of anticoagulant medications in patients with atrial fibrillation. We found an excellent correlation between devices. However, assessment of the PT/INR test capability to guide anticoagulant treatment in patients with a prosthetic cardiac valve (particularly in the mitral position) or plasma transfusion in severe trauma patients remains mandatory [3,5,27].

In this work, we confirmed, in real-life situations, that the i-STAT Alininity technical performances reported by the manufacturer remained applicable and that the analytical performances were close to those of the central laboratory instruments. More importantly, this POC device had good clinical performances providing sufficiently reliable results for therapeutic decision making. Thus, our results showed that i-STAT Alinity can meet the critical therapeutic needs of patients in the ED or ICU and that the device was as easy to use, as testing can be performed by nurses or physicians searching for a POC device with a wide range of analyses.

We must acknowledge some limitations to this study. First, we did not assess the linearity, interference and reproducibility of the i-STAT Alinity analyzer. However, previous studies have reported good linearity (five-level calibration range) and reproducibility (one run per day of each control level for 14 days) for all the parameters except for PT and INR [1,28]. Second, we did not comply with the Clinical and Laboratory Standards Institute (CLSI) EP15-A3 protocols to perform our imprecision study, since they are not applicable for POC devices. However, we followed a rigorous methodology similar to those previously used for POC device evaluations [1,4]. Third, we used another blood gas analyzer (ABL 90 Flex Plus) in the comparison study for most of parameters that could have underestimated biases. Fourth, we did not perform stress testing on the device at extreme environmental temperatures or humidity. In our study, the devices were localized in a temperature-controlled environment, and the test cartridges were all stored in temperature-controlled containers. Consequently, our results are not generalizable for use in the provision of care during emergency medical services calls, critical care transport or disaster relief missions. Indeed, others recently reported that the analytical performances of the i-STAT Alinity POC system changed in an environment with extreme temperatures [29]. Additionally, we performed the PT and INR tests on a neutral purge tube, which could have flawed the results of the PT and INR measurements due to blood dilution and/or the activation of coagulation in the tube before the test. Last, we did not assess the operator experiences in our study; nevertheless, the i-STAT Alinity POC system has been previously reported to be ergonomic and easy to use, and most of its user have been satisfied [1].

## 5. Conclusions

The i-STAT Alinity system, a handheld, portable, POC blood testing device, has shown adequate imprecision and comparable accuracy to the existing laboratory methods for pH, Pco_2_, base excess, bicarbonate, lactate, electrolytes, glucose, urea and hemoglobin measurements. However, the analytical imprecision did not meet the international recommendations for central laboratory devices showing bias in the measurements of Po_2_, creatinine, hematocrit and PT/INR. The i-STAT Alinity system could be an effective POC testing device for triage at patients’ bedsides in the ED or in the ICU; nevertheless, clinicians should interpret the results of the Po_2_, creatinine, hematocrit and PT/INR with caution. Further studies are mandatory to assess the capability of such a device to improve patient management and prognosis in real-life settings.

## Figures and Tables

**Figure 1 diagnostics-13-00297-f001:**
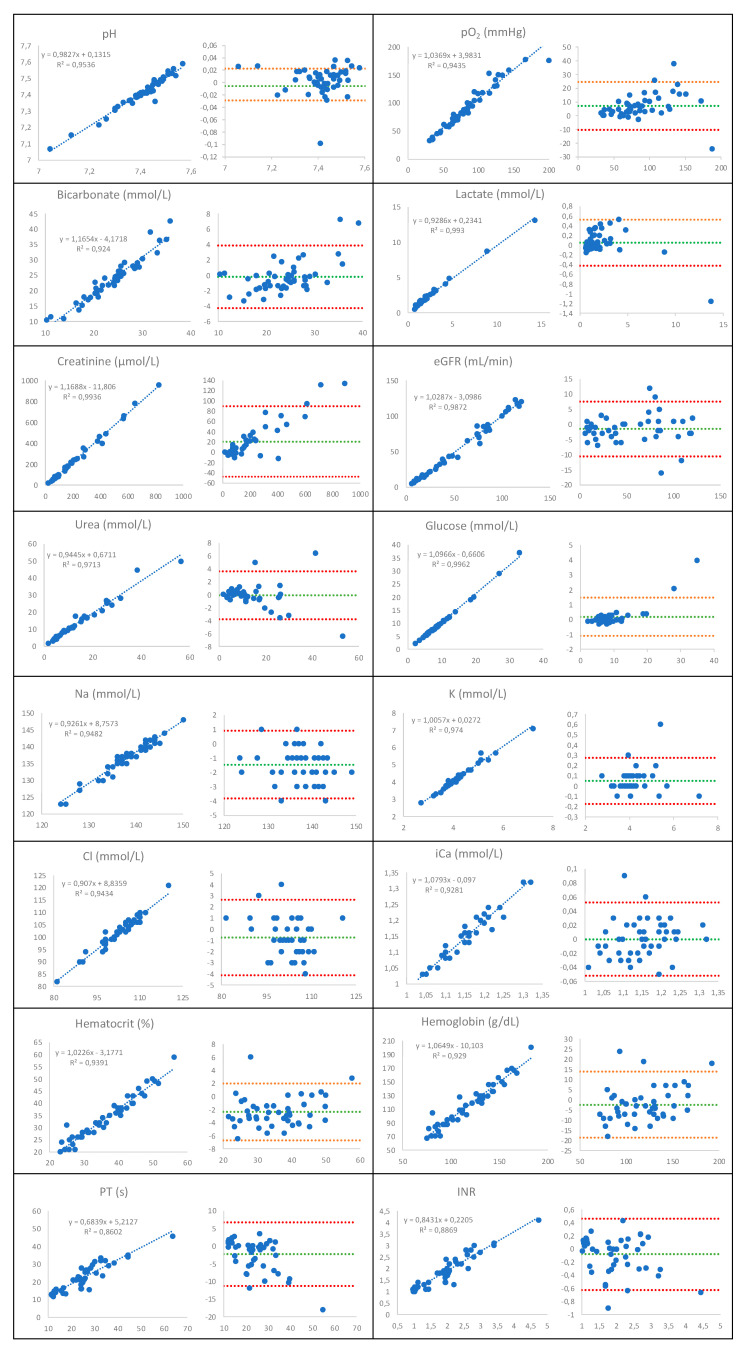
Passing–Bablok regressions of pH, partial pressure of oxygen (Po2), bicarbonate, lactate, glucose, sodium (Na), potassium (K), chloride (Cl) and ionized calcium (iCa), hematocrit and hemoglobin on the iSTAT Alinity (*y* axis) and ABL 90 Flex Plus (*x* axis) analyzers; creatinine, estimated glomerular filtration rate (eGFR) and urea on the iSTAT Alinity (*y* axis) and Cobas 8000 (*x* axis) analyzers and prothrombin time (PT) and international normalized ratio (INR) on the iSTAT Alinity (*y* axis) and ACL TOP analyzers (*x* axis). Bland–Altman plots of differences against means for patient samples (N = 49, except for creatinine and urea: N = 45, and PT and INR: N = 46) measurements with the iSTAT Alinity and central laboratories’ analyzers represented with the mean (dashed green line) and limits of agreement (dashed red lines) of the bias.

**Table 1 diagnostics-13-00297-t001:** Imprecision study results.

Cartridge	Control		Analyte	Unit	N	Range of Results	Mean	SD (%)	CV (%)	CV Goals (%) Ricos [13,14] (Desirable) /SFBC	Abbott CV (%)
CG4+	L1	7.049	pH	-	20	6.999–7.099	7.065	0.004	0.05	0.1/-	0.08
	L3	7.643	pH	-	20	7.593–7.693	7.649	0.004	0.06	0.1/-	0.04
	L1	62.1	Pco_2_	mmHg	20	54.6–69.6	59.0	1.1	1.8	2.4/3.8	2.5
	L3	23.1	Pco_2_	mmHg	20	16.8–29.4	23.2	0.5	2.2	2.4/4.5	2.0
	L1	85	Po_2_	mmHg	20	70–100	85	1.7	2.0	-/1.5	4.79
	L3	148	Po_2_	mmHg	20	126–170	136	2.3	1.7	-/1.5	4.10
	L1	6.68	Lactate	mmol/L	20	5.77–7.58	6.73	0.05	0.8	13.6/3.8	1.21
	L3	0.69	Lactate	mmol/L	20	0.19–1.19	0.74	0.01	2.0	13.6/3.8	3.27
CHEM8+	L1	124	Na	mmol/L	20	119–128	123	0.4	0.3	0.3/1	0.4
	L3	159	Na	mmol/L	20	153–165	159	0.8	0.5	0.3/0.7	0.3
	L1	3.0	K	mmol/L	20	2.7–3.3	2.9	0.04	1.4	2.3/1.5	1.3
	L3	6.1	K	mmol/L	20	5.6–6.7	6.2	0.05	0.8	2.3/1.2	0.6
	L1	73	Cl	mmol/L	20	68–78	74	0.5	0.6	0.6/1.2	0.7
	L3	115	Cl	mmol/L	20	106–123	115	0.7	0.6	0.6/1.2	0.5
	L1	0.89	iCa	mmol/L	20	0.81–0.97	0.90	0.007	0.8	0.9/1.2	1.1
	L3	1.57	iCa	mmol/L	20	1.45–1.68	1.58	0.013	0.8	0.9/1.2	1.4
	L1	14.2	Glucose	mmol/L	20	11.8–16.5	14.4	0.06	0.4	2.8/1.2	1.6
	L3	2.1	Glucose	mmol/L	20	1.6–2.6	2.1	0.04	2.1	2.8/2.4	0.8
	L1	20.3	Urea	mmol/L	20	17.9–23.2	20.4	0.44	2,1	6.1/1.9	1.4
	L3	2.1	Urea	mmol/L	20	1.1–2.9	2.1	0.09	4.3	6.1/4.5	8.2
	L1	309	Cr	µmol/L	20	239–380	333	6.5	2.0	3.0/3.4	3.0
	L3	44	Cr	µmol/L	20	0–80	35	1.8	5.2	**3.0/4.5**	**4.8**
	L1	22	Ht	%	20	19–25	22	0.2	1.0	1.35/-	1.5
	L3	56	Ht	%	20	53–60	55	0.5	0.9	1.35/-	1.0
PT/INR	L1	16.5	PT	s	20	12.6–20.4	14.6	0.38	2.6	**2.0/-**	/
	L2	28.7	PT	s	20	21.6–35.8	25.6	1.60	6.3	**2.0/-**	/
	L1	1.4	INR	-	20	1.1–1.7	1.2	0.05	3.7	-/-	**4.5**
	L2	2.5	INR	-	20	1.9–3.1	2.2	0.08	3.7	-/-	**6.9**

Abbreviations: Cl = chlorine, Cr = creatinine, CV = coefficient of variation, Ht = hematocrit, iCa = ionized calcium, INR = international normalized ratio, K = potassium, Na = sodium, Pco_2_ = partial pressure of carbon dioxide, Po_2_ = partial pressure of oxygen, PT = prothrombin time, SD = standard deviation and SFBC = Societe Française de Biologie Clinique (French Society of Clinical Biology). Reference values of CVs under the CV reported in the present study are in bold.

**Table 2 diagnostics-13-00297-t002:** Results of the Passing–Bablok regression and Bland–Altman analyses for the iSTAT Alinity measurements compared to the existing laboratory instruments.

		Passing–Bablok Regression			Bland Altmann				
Analyte	Reference	N	Slope	y Intercept	R^2^	Bias (%)	Bias	Lower Limit of Agreement	Upper Limit of Agreement
pH	ABL 90	49	0.983	0.131	0.954	0.05	0.003	−0.029	0.022
Po_2_	ABL 90	49	1.037	3.983	0.944	7.9	7	−10	25
Lactate	ABL 90	49	0.929	0.234	0.993	3.3	0.05	−0.43	0.53
Na	ABL 90	49	0.926	8.757	0.948	−1.1	−1	−4	1
K	ABL 90	49	1.006	0.027	0.974	1.2	0.1	−0.2	0.3
Cl	ABL 90	49	0.907	8.836	0.943	−0.7	−1	−4	3
iCa	ABL 90	49	1.079	−0.097	0.928	−0.04	0	−0.05	0.05
Glu	ABL 90	49	1.097	−0.661	0.996	1.1	0.2	−1.1	1.5
Urea	COBAS	45	0.944	0.671	0.971	0.8	−0.1	−3.7	3.6
Cr	COBAS	45	1.169	−11.806	0.994	5.4	21	−48	89
Ht	ABL 90	49	1.023	−3.177	0.939	−7.3	−2	−7	2
Hb	ABL 90	49	1.065	−10.103	0.929	−2.7	−2.4	−18.7	13.9
PT	ACL TOP	46	0.684	5.213	0.860	−6.6	−2.2	−11.2	6.8
INR	ACL TOP	46	0.843	0.220	0.897	−3.0	−0.1	−0.6	0.5

Abbreviations: Cl = chlorine, Cr = creatinine, Glu = glucose, Hb = hemoglobin, Ht = hematocrit, iCa = ionized calcium, INR = international normalized ratio, K = potassium, Na = sodium, Po_2_ = partial pressure of oxygen, PT = prothrombin time, R^2^ = coefficient of correlation. Units: mmHg (Po_2_), mmoL/L (lactate, Na, K, Cl, iCa, Glu, urea), µmoL/L (creatinine), % (Ht), g/dL (Hb) and seconds (PT).

**Table 3 diagnostics-13-00297-t003:** Concordance of Chronic Kidney Disease (CKD) staging induced by creatinine measured by the i-STAT Alinity system (Abbott laboratories, Chicago, IL, USA) and Cobas 8000 instrument using a c702 module (Roche Diagnostic, Meylan, France) in 45 patients.

	i-STAT Alinity							
	CKD Stage (eGFR)	I	II	IIIa	IIIb	IV	V	Total
COBAS 8000	I (>90 mL/min)	9	0	0	0	0	0	9
	II (60–90 mL/min)	0	12	0	0	0	0	12
	IIIa (60–45 mL/min)	0	0	0	3	0	0	3
	IIIb (45–30 mL/min)	0	0	0	4	2	0	6
	IV (30–15 mL/min)	0	0	0	0	5	2	7
	V (<15 mL/min)	0	0	0	0	0	8	8
	Total	9	12	0	7	7	10	45

Abbreviations: CKD = chronic kidney disease. CKD stages according to the Kidney Disease: Improving Global Outcomes (KDIGO) criteria [10] and eGFR = estimated glomerular filtration rate. Concordant (grey) and discordant (orange) measurments.

**Table 4 diagnostics-13-00297-t004:** Concordance with antivitamin K drug regimen changes induced by the international normalized ratio (INR) measurements by the i-STAT Alinity system (Abbott laboratories, Chicago, IL, USA) and ACL TOP 700 analyzer (Werfen, Barcelona, Spain) in 46 patients.

	i-STAT Alinity				
Change in Antivitamin K Drug Dosage Regimen		Dose Increase (INR < 2)	Same Dose (INR = 2–3)	Dose Decrease (INR > 3)	Total
ACL TOP 700	Dose increase (INR < 2)	26	0	0	26
	Same dose (INR = 2–3)	3	14	0	17
	Dose decrease (INR > 3)	0	1	2	3
	Total	29	15	2	45

Abbreviations: INR = international normalized ratio. INR target according to the 2020 European Society of Cardiology (ESC) guidelines for the diagnosis and management of atrial fibrillation [11]. Concordant (grey) and discordant (orange) measurments.

## Data Availability

The authors consent to share the collected data with others. The raw data supporting the conclusions of this article will be made available by the authors, without undue reservation. The data will be available immediately after the main publication indefinitely.

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
