# Peer review of "Analytical Performances of the Novel i-STAT Alinity Point-of-Care Analyzer"

_diagnostics, 2023, doi:10.3390/diagnostics13020297_

Round 1

Reviewer 1 Report

In this paper, the authors aim to assess the analytical performances of the i-STAT Alinity system. The authors conducted an analytical perfomances study with the i-STAT Alinity device using cartridges CG4+ (pH, Pco2, Po2, lactate, bicarbonate, base excess), CHEM8+ (Na, K, Cl, ionized Ca, urea, creatinine, glucose, hematocrit and hemoglobin) and PT/INR (prothrombin time and international normalized ratio). They assessed 21 imprecision and compared the results to those obtained on existing instruments in the central laboratory.

My main concern about this work is the lack of novelty. There is no research element here. They test a commercial product and they find that it works! The opposite would have been alarming but these results were widely expected. Moreover, as they mention in lines 243 - 249, other previous works already focused on testing the device so this work is not the first example of this kind of test.

The paper also needs to be reworked to stress the research element. First, they must provide an extensive literature review on similar commercial devices and explain why they are testing this specific model. 

Additionally, it would be interesting to do some stress testing on these devices. For example, what do I get if I run a test with an environmental temperature of 10 celsius degree or 30 degrees? What if we change the environmental humidity? Are the results still valid? Is a POC device more robust against these sources of error with respect to standard equipment? 

However, the dataset presented and the effort to get that data must be acknowledged. I would suggest replotting all the graphs in Figure 1 using more professional software such as origin or Matlab. 

Finally, the dataset should be shared with the community for open science practices. 

Author Response

In this paper, the authors aim to assess the analytical performances of the i-STAT Alinity system. The authors conducted an analytical perfomances study with the i-STAT Alinity device using cartridges CG4+ (pH, Pco2, Po2, lactate, bicarbonate, base excess), CHEM8+ (Na, K, Cl, ionized Ca, urea, creatinine, glucose, hematocrit and hemoglobin) and PT/INR (prothrombin time and international normalized ratio). They assessed 21 imprecision and compared the results to those obtained on existing instruments in the central laboratory.

The authors thank the reviewer for their remarks and for their thorough comments

My main concern about this work is the lack of novelty. There is no research element here. They test a commercial product and they find that it works! The opposite would have been alarming but these results were widely expected. Moreover, as they mention in lines 243 - 249, other previous works already focused on testing the device so this work is not the first example of this kind of test.

We thank the reviewer for raising this concern. It seems very important to us to emphasize that there are differences between technical and clinical requirements that can be expected from a device. As correctly pointed by the reviewer POC devices are commercial product that meet technical requirements. The purposes of analytical performance studies are to confirm technical requirements remain true in real-life setting (the reference methods are those in the central lab) and to assess the impact of imprecision on clinical decision including concordance of threshold value of therapeutic decision making. Especially, for POC devices, the analytical performance could be impacted by the fact they are used by physicians, nurses or midwives, not by trained laboratory technicians and biologists.

In addition, we would like to underline there are many POC systems available on the market, released by different manufacturers. Among manufacturers, Abbott offers several devices with different technical characteristics but very similar names: i-STAT-1, i-STAT Alinity, i-STAT Wireless. As a matter of fact, there are very few papers evaluating the POC device named “i-STAT Alinity” (only three are indexed in medline to our knowledge). Thus, our work was the first to assess the analytical performance of the i-STAT Alinity for TP and INR, the first to assess a large set of parameters including both biochemical and hematological tests, and to report that clinicians can use this device for eGFR evaluation and VKA treatment monitoring. We amended the introduction accordingly. Please see lines 48-62, p2

The paper also needs to be reworked to stress the research element. First, they must provide an extensive literature review on similar commercial devices and explain why they are testing this specific model.

As suggested by the reviewer we have made a deep review of the literature on similar commercial devices. This review clearly shows that the i-STAT Alinity is one of the most complete apparatuses which could perform using the same device blood gas analyses, ionic disturbances monitoring, kidney function evaluation, anti-vitamin K follow-up. Moreover, the analytical performances of the i-STAT were in the range of other systems. Please see Table S1 included in the supplementary material. This precision has been also added in the discussion. Please see lines 260-270, pp8-9.

Additionally, it would be interesting to do some stress testing on these devices. For example, what do I get if I run a test with an environmental temperature of 10 celsius degree or 30 degrees? What if we change the environmental humidity? Are the results still valid? Is a POC device more robust against these sources of error with respect to standard equipment? 

We agree with the reviewer, it is a very important issue to assess the analytical performances of such a device in various environmental temperature and humidity. However, our study was focused on POC testing in hospital setting, especially, in ICU and ED, not in the provision of care during EMS calls or critical care transport. We acknowledge this limit in the last paragraph of the discussion, and we cited the review of Füzéry et al. reporting the i-STAT Alinity has questionable effectiveness in extreme temperatures. However, in hospital settings, devices were localized in a temperature control and shock resistant environment. Test cartridges were all stored in temperature-controlled containers

However, the dataset presented and the effort to get that data must be acknowledged. I would suggest replotting all the graphs in Figure 1 using more professional software such as origin or Matlab.

We apologize to the reviewer but we do not have this kind of software. However, we replotted all the graphs in the figure 1 to improve its design. Please see Figure 1.

Finally, the dataset should be shared with the community for open science practices. 

As requested by the reviewer, we have changed the data availability statement to facilitate the sharing of the data that supports the conclusion of our work.

Reviewer 2 Report

In this work, the authors performed an objective comparison of the analytical capabilities of a commercial POC device (i-STAT Alinity) against central lab equipment. The number of patients, the methodology of sample process, and the statistical treatment of the information seems to be correct. The conclusions of the work provide valuable information about this particular POC device, but also emphasize the need for more studies like this. The work is well focused within the scope of the journal and of great interest for the clinical analysis community. The paper is well written in terms of both english grammar and scientific language. Figures and tables are correct  in number and content. I recommend the acceptance of the work after a few minor corrections. 

Line 47: the dots should be replaced by “among others” or  similar.

Lines 49–54: The sentence should be rephrased as far as it is difficult to understand on its current form.

Lines 214–215: considering that 4 patients received a non-ideal  treatment (roughly 10%), I’d avoid to describe the agreement for the CKD  staging as “excellent”.

Author Response

In this work, the authors performed an objective comparison of the analytical capabilities of a commercial POC device (i-STAT Alinity) against central lab equipment. The number of patients, the methodology of sample process, and the statistical treatment of the information seems to be correct. The conclusions of the work provide valuable information about this particular POC device, but also emphasize the need for more studies like this. The work is well focused within the scope of the journal and of great interest for the clinical analysis community. The paper is well written in terms of both english grammar and scientific language. Figures and tables are correct  in number and content. I recommend the acceptance of the work after a few minor corrections. 

First, we thank the reviewer for their interest to this work and their helpful observations.

Line 47: the dots should be replaced by “among others” or  similar.

As suggested by the reviewer, we changed the dots by among others. Please see line 47, p2.

Lines 49–54: The sentence should be rephrased as far as it is difficult to understand on its current form.

As correctly pointed by the reviewer, a paragraph in the introduction was unclear. The sentences of this paragraph were entirely rephrased. Please see lines 48-54, p2.

Lines 214–215: considering that 4 patients received a non-ideal  treatment (roughly 10%), I’d avoid to describe the agreement for the CKD  staging as “excellent”.

We completely agree with the reviewer, the term "excellent" has been removed. Please see line 236, p8.

Round 2

Reviewer 1 Report

The authors have now addressed my concerns. They have now improved the paper and explained clearly why this work is relevant. They also plan to share the whole dataset with the community, which is very useful in my opinion. 

Please, improve the discussion about technical and clinical performance. Also, please discuss how the mismatch between your findings and stated performance can affect medical decisions. Add details about how this work has an impact on society. 

Author Response

The authors have now addressed my concerns. They have now improved the paper and explained clearly why this work is relevant. They also plan to share the whole dataset with the community, which is very useful in my opinion.

The authors thank again the reviewer for their constructive remarks.

Please, improve the discussion about technical and clinical performance. Also, please discuss how the mismatch between your findings and stated performance can affect medical decisions. Add details about how this work has an impact on society.

We complied with the reviewer’s recommendations and added a paragraph in the discussion. Please see line 318-331 pp 9-10.